# A Synthetic Approach for Biosynthesis of Miquelianin and Scutellarin A in *Escherichia coli*

**Ramesh Prasad Pandey** [1,2,†] , **Ha Young Jung** [1,†] , **Prakash Parajuli** [1] ,
**Thi Huyen Trang Nguyen** [1] , **Puspalata Bashyal** [1] and **Jae Kyung Sohng** [1,2,*]

[1] Department of Life Science and Biochemical Engineering, Sun Moon University, 70 Sunmoon-ro 221, Tangjeong-myeon, Asan-si, Chungnam 31460, Korea; ramesh.pandey25@gmail.com (R.P.P.); gkdud0328@naver.com (H.Y.J.); parajuli1985@gmail.com (P.P.); nguyenhuyentrang0512@gmail.com (T.H.T.N.); latabashyall@gmail.com (P.B.)

[2] Department of Pharmaceutical Engineering and Biotechnology, Sun Moon University, 70 Sunmoon-ro 221, Tangjeong-myeon, Asan-si, Chungnam 31460, Korea

* Correspondence: sohng@sunmoon.ac.kr; Tel.: +82-(41)-530-2246; Fax: +82-(41)-544-2919

† These authors equally contributed.

**Abstract:** Grapevine (*Vitis vinifera*) glycucuronosyltransferase (VvGT5) specifically catalyzes flavonol-3-*O*-glucuronosylation and the blue flowers of *Veronica persica* (Lamiales, Scrophulariaceae) uridine diphosphate (UDP)-dependent glycosyltransferase (UGT88D8) as flavonoid 7-*O*-specific glucuronosyltransferases, were chosen, codon optimized, and employed to synthesize the high valued flavonoids glucuronoids, miquelianin and scutellarin A in *Escherichia coli*. A single vector system was constructed to overexpress entire UDP-glucuronic acid biosynthesis pathway genes, along with a glucokinase gene in *Escherichia coli* BL21 (DE3). The newly generated *E. coli* BL21 (DE3) piBR181-glk.pgm2.galU.ugd.UGT88D8 strain produced 12 mg/L (28 µmol/L) of scutellarin A from apigenin, representing only 14% of maximum conversion percentage. Similarly, the strain *E. coli* BL21 (DE3) piBR181-glk.pgm2.galU.ugd.VvGT5 produced 30 mg/L (62 µmol/L) of miquelianin, representing a 31% conversion of quercetin. This production profile is a good starting point for further host engineering, and for production of respective compounds.

**Keywords:** flavonoid *O*-glucuronosyltransferase; multi-monocistronic vector; UDP-glucuronic acid; miquelianin; scutellarin A

## 1. Introduction

Flavonoids have always been recognized as a promising factor in human health, as they refer to various biological responses, such as an inherent ability to modify the body's reaction to allergens, viruses, and carcinogens. Experimental pieces of evidences have always been positive in terms of their activities as anti-allergens, anti-inflammatory, anti-microbial, anti-cancer, etc. [1–3]. Flavonoids cover a large group of plant secondary metabolites with a profound research interest in various biological and therapeutic applications, since its genetic and biosynthetic investigations have been well-documented over the decades [4]. These aromatic molecules are derived from *p*-coumaroyl-CoA and malonyl-CoA. They cover a diverse class of natural products, including; the chalcones, flavones, isoflavones, flavonols, flavandiols, anthocyanins, and condensed tannins [5,6]. Flavonols (quercetin, kaempferol, myricetin, isorhamnetin) represent their diversity, and stem from the 3-OH phenolic group. They are present in a wide variety of fruits and vegetables (apples, berries, onions, broccoli, grapes, flowers and bark), including botanicals [6]. They are well-known for their various health-promoting effects, which include antioxidant, anti-inflammatory, anti-angiogenic, anti-proliferative, and neuro-pharmacological

properties [1–3]. However, flavones (apigenin, luteolin, chrysin, baicalein) have recently been generating enormous public and scientific interest due to their beneficial effects against atherosclerosis, osteoporosis, diabetes mellitus, and certain cancers [7–9]. Interestingly, both classes of flavonoids act as inhibitors of cytochrome (CYP) P450 enzymes, such as CYP2C9 and CYP3A4, enzymes that help in the metabolism of xenobiotics in the human body [10,11].

Thus, polyphenols are worth intaking naturally as they have substantial health benefits; however, there are limitations in direct consumption. The limitations could be rooted in their water solubility, limited bioavailability or xenobiotic neutralizations, which challenges the necessity of their modifications by glycosylation/hydroxylation (to enhance solubility), and methylation (to increase lipophilicity, cell recognition) [12–14]. Nevertheless, conjugation of carbohydrate appendages in such promising metabolites has always improved pharmaceutical values [15]. The glucuronic acid itself is often linked to the xenobiotic metabolism in the human body, and the substances resulting from glucuronidation are typically much more water-soluble [16]. In plants and animals, glucuronic acid is considered to be a precursor of vitamin C [17]. Thus, producing plant polyphenols conjugated with sugars such as glucuronic acid could help to develop such molecules as prospective food additives.

Biological synthesis of regio- and stereo-selective phytochemicals using an industrially important *Escherichia coli* system has been widely studied in current research trends [14]. Most of the bacteria, including *E. coli*, can produce nucleotide sugars, which are building blocks of bacterial polysaccharides and have been well-studied for various reasons [14,18]. These activated nucleotide diphosphate sugars are utilized as sugar donors by glycosyltransferase (UGTs), which are central players in decorating these valuable natural plant products with sugar moieties [19]. Although sugars like UDP-glucose, UDP-galactose, and UDP-glucuronic acid are common nucleotide diphosphate sugars present in *E. coli*, engineering or overexpression of the pathway-specific genes increase their availability for the biotransformation.

The current research employs *Vitis vinifera* glucuronosyltransferase; *VvGT5* catalyzes the 3-*O*-specific glucuronic acid conjugation, and flavonoid 7-*O*-specific glucuronosyltransferase, *UGT88D8* from the blue flowers of *Veronica persica* to produce miquelianin and scutellarin A in *E. coli*. Both plant UGTs were codon optimized, synthesized, and functionally expressed in *E. coli*. Moreover, both genes were incorporated in single vector multi-monocistronic systems, overexpressing UDP-glucoronic pathway genes to develop two independent systems for regiospecific glycosylation.

## 2. Materials and Methods

### 2.1. Bacterial Strains and Culture Conditions

*E. coli* XL-1 Blue (Stratagene, CA, USA) was used for plasmid cloning and propagation, whereas *E. coli* BL21 (DE3) was used for expression of the proteins and flavonoid biotransformation. All strains, vectors, and plasmids used in this study are listed in Table 1 and the construction of the mono-cistronic vector system is described in (Figure 1). Cloning and manipulations of DNA were performed according to the standard procedures [20]. A previously constructed expression vector piBR181 [21] was used for cloning and expression. The pGEM-T® easy vector system (Promega, Madison, WI, USA) was used as a vector for cloning polymerase chain reaction (PCR) fragments and sequencing. *E. coli* BL21 (DE3) Δ*pgi*Δ*zwf* Δ*ushA* mutant harboring the UDP-glucose biosynthetic pathway was used as a final strain for biotransformation. *E. coli* strains were routinely cultured in Luria-Bertani (LB) broth or on LB agar plates, supplemented with the appropriate antibiotics (ampicillin 100 μg/mL, kanamycin 50 μg/mL) when necessary. Authentic apigenin and quercetin were purchased from Sigma-Aldrich (St. Louis, MO, USA), and the rest of the chemicals used in this study were high-grade and purchased from commercially-available sources.

**Figure 1.** Schematic diagram of the uridine diphosphate (UDP)-glucuronic acid biosynthesis pathway and its utilization by UGT88D8 and VvGT5 to transfer glucuronic moiety to 7-OH position of apigenin and 3-OH position of quercetin, respectively.

**Table 1.** List of primers used in this study.

| Primers | Oligonucleotide Sequences (5′-3′) | Total PCR Size | Restriction Sites |
| --- | --- | --- | --- |
| glk-F | TCTAGAATGGAAATTGTTGCGATTGACATCGGT | 984 bp | *Xba*I |
| glk-R | AAGCTTTTATTCAACTTCAGAATATTTGTTGGC | 984 bp | *Hind*III |
| pgm2-F | TCTAGAATGAGCTGGAGAACGAGCTATGAACGC | 1749 bp | *Xba*I |
| pgm2-R | AAGCTTTTACGAATTTGAGGTCGCTTTTACAAT | 1749 bp | *Hind*III |
| galU-F | TCTAGAATGGCTGCCATTAATACGAAAGTCAAA | 908 bp | *Xba*I |
| galU-R | AAGCTTTTACTTCTTAATGCCCATCTCTTCTTC | 908 bp | *Hind*III |
| ugd-F | TCTAGAATGAAAATCACCATTTCCGGTACTGGC | 1167 bp | *Xba*I |
| ugd-R | AAGCTTTTAGTCGCTGCCAAAGAGATCGCGGGT | 1167 bp | *Hind*III |
| UGT88D8-F | TCTAGAATGGAAGACACGATTATTCTGTATGCC | | *Xba*I |
| UGT88D8-R | AAGCTTTGATGAACTTATCCAGATCCACCACGC | | *Hind*III |
| VvGT5-F | TCTAGAATGACCACGACCGCCTCCTCAATGGAT | | *Xba*I |
| VvGT5-R | AAGCTTCGTGTCCAGCGGCAGTTTAGAGGTGGT | | *Hind*III |

### 2.2. Cloning and Expression of Recombinant Proteins

Restriction enzymes, *LaTaq* polymerase, and T4 DNA ligase were purchased from Takara Bio Inc. (Shiga, Japan). UDP-glycosyltransferases, *VvGT5* (grapevine; *Vitis vinifera* UDP-glucuronic acid: flavonol 3-*O*-glucuronosyltransferase) and *UGT88D8* (the blue flowers of *Veronica persica* poiret; *Lamiales, Scrophulariaceae* structurally flavonoid 7-*O*-specific glucuronosyltransferases (F7GAT)) were codon optimized, synthesized, cloned and expressed in restriction site dependent multi-mono-cistronic vector, piBR181. Oligonucleotide primers were synthesized from Genotech (South Korea). All PCR size and primers used in this study are described in Table 2. To increase the pool of UDP-glucuronic acid, its pathway specific genes were individually cloned and assembled together in the piBR181 vector. The UDP-glucuronic acid sugar pathway genes glucokinase (*glk*: from *Zymomonas mobilisi*, GenBank ID: AE008692.2), a vector pVWEx1-*glf*-*glk* provided by Dr. Stephanie Bringer-Meyer (Institut für Bio- und Geowissenschaften, IBG-1: Biotechnologie, Forschungszentrum Jülich, Jülich,

Germany), phosphoglucomutase (*pgm2*; from *Bacillus licheniformis* DSM 13, GenBank accession No. YP_006712377.1), glucose 1-phosphate uridylyltransferase (*galU*; from *E. coli* K-12, GenBank accession No. CP001509.3), and UDP-glucose 6-dehydrogenase (*ugd*; GenBank accession No. YP_490271.1 from *E. coli* BL21(DE3) were selected for overexpression. All these genes were amplified from the genomic DNA of respective strains using a pair of primer that had *Xba*I/*Hin*dIII restriction sites, as shown in Table 1. Finally, the sugar cassette was assembled with codon optimized regiospecific glycurunosyltransferases (*UGT88D8* or *VvGT5*) separately, as the physical scheme explains in Figure 1. These two plasmids were used for enhanced biosynthesis of two molecules. In all cases, construction of each recombinant and assembly of each pathway gene were verified with restriction digestion, PCR, and direct nucleotide sequencing of PCR-amplified products of the respective genes in the recombinant plasmid. Each recombinant plasmid containing UDP-glucuronic acid sugar pathway genes was transformed into *E. coli* BL21 (DE3) with the heat shock transformation technique. Later, the restriction-digestion-confirmed single transformant was cultured for further studies. Sodium dodecyl sulfate-polyacrylamide gel electrophoresis (SDS-PAGE) analysis was performed to check the expression of each gene and was compared with the standard protein marker. To check the expression of each gene, we followed the same methodology as described previously [22] with the same concentration of IPTG. However, the protein expression was carried out at 15 °C.

**Table 2.** List of bacterial strains and vector maps used in this study.

| Strains/Plasmids | Description | Source/Reference |
|---|---|---|
| *Escherichia coli* XL-1 Blue (MRF′) | General cloning host | Stratagene, CA USA |
| *E. coli* BL21 (DE3) | *ompT hsdT hsdS* (r$_B$-m$_B$-) gal (DE3) | Novagen, Madison, WI, USA |
| *E. coli* BL21 (DE3)/piBR181-UGT88D8 | *E. coli* BL21 (DE3) carrying piBR181-UGT88D | This study |
| *E. coli* BL21 (DE3)/piBR181-VvGT5 | *E. coli* BL21 (DE3) carrying piBR181-VvGT5 | This study |
| *E. coli* BL21 (DE3)/piBR181-glk.pgm2.ugd.UGT88D8 | *E. coli* BL21 (DE3) carrying piBR181.glk, pgm2, galU, ugd and UGT88D8 | This study |
| *E. coli* BL21 (DE3)/piBR181-glk.pgm2.ugd.VvGT5 | *E. coli* BL21 (DE3) carrying piBR181.glk, pgm2, galU, ugd, and VvGT5 | This study |
| **Plasmid and Vectors** | | |
| pGEM-T® easy vector | General cloning vector, T7, and SP6 promoters, f1 ori, Amp$^r$ | Promega, Madison, WI, USA |
| pIBR181 | Mono-cistronic vector | [21] |
| piBR181 piBR181-UGT88D8 | The piBR181 vector carrying *UGT88D8*, codon optimized flavonoid 3-*O*-glucuronosyltransferase from blue flowers of *Veronica persica* | This study |
| piBR181 piBR181-VvGT5 | The piBR181 vector carrying *VvGT5*, codon optimized flavonoid 7-*O*-glucuronosyltransferase from grapevine (*Vitis vinifera*) | This study |
| piBR181-glk.pgm2.galU.ugd. UGT88D8 | The piBR181 vector carrying *glk* from *Zymomonas mobilis*, *pgm2* from *Bacillus licheniformis*, *galU* and *ugd* from *E. coli* K12 and *UGT88D8* from blue flowers of *Veronica persica* | This study |
| piBR181-glk.pgm2.galU.ugd. VvGT5 | The piBR181 vector carrying *glk* from *Zymomonas mobilis*, *pgm2 Bacillus licheniformis*, *galU* and *ugd* from *E. coli* K12 and *VvGT5* from grapevine (*Vitis vinifera*) | This study |
| pVWEx1-glf-glk | pVWEx1 carrying *glk* | [23] |

*2.3. Construction of Sugar Cassettes*

All genes cloned from different organism sources were restriction digested with *Xba*I/*Hin*dIII after routine manipulation in pGEM-T Easy vector, and individually cloned into the expression vector piBR181 at the restriction site of *Spe*I/*Hin*dIII. *Spe*I and *Xba*I are cohesive enzymes as they generate

similar over-hangs. Thus, ligation of these two fragments is possible, but after ligation a 'scar' is formed, which replaces the restriction sites. In the vector backbone, *Bam*HI/*Xho*I site remains, which can be applied for further cloning of other genetic parts digested with the *Bgl*II and *Xho*I enzymes (includes a set; an RBS, a T7 promoter, a target gene, and a transcription terminator). Similarly, *Bgl*II and *Bam*HI also generate cohesive ends and forms a scar after ligation. This means, in each insertion of a circuit part, there is the formation of a scar. This restricts further cuts by the same restriction endonucleases, leaving a new cloning site at *Bam*HI/*Xho*I in the vector. Likewise, the cloning of each sugar pathway gene, including UGTs, completed the two cassettes of glucuronosylation system.

## 2.4. Whole-Cell Biotransformation and Flavonoid Extraction

*E. coli* strains were pre-cultured in 3 mL of LB liquid medium with antibiotics and incubated at 37 °C in 200 rpm overnight. Five-hundred microliters of inoculum was transferred into 50 mL of Luria-Bertani (LB) liquid medium with appropriate antibiotics and cultured at 37 °C until the optical density at $OD_{600nm}$ reached 0.5–0.7. The culture was induced by adding isopropyl-$\beta$-D-thiogalactopyranoside (IPTG) with a final concentration of 0.5 mmol/L and incubated at 15 °C for protein expression, and to increase cell biomass. The following day, each culture was supplemented with specific flavonoids; apigenin in *E. coli* harboring UGT88D8 to produce scutellarin A, and quercetin in *E. coli* harboring VvGT5 to produce miquelianin. Then, the flasks were incubated at 15 °C for 60–72 h for biotransformation. Each culture was extracted with an equal volume of ethyl acetate. The organic layer was collected and concentrated to dryness through the evaporation of excess solvent, followed by the addition of methanol to dissolve remaining products. The extracts were subjected to high performance liquid chromatography (HPLC) and liquid chromatography quadrupole time-of-flight-electrospray ionization mass spectrometry (LC-QTOF-ESI/MS) analyses.

Following the confirmation of products, we performed the bio-catalysis reaction with the strains containing sugar cassettes, including UGTs. Equal quantities (200 μL) having the same cell density (0.7) inoculums were transferred into 50 mL *E. coli* flasks containing LB medium. Identical conditions were applied for IPTG induction, incubation temperature, and shaking of the culture. *E. coli* BL21 (DE3) Δ*pgi*Δ*zwf*Δ*ushA* piBR181-glk.pgm2.galU.ugd.UGT88D8 was fed with apigenin while *E. coli* BL21 (DE3) Δ*pgi*Δ*zwf*Δ*ushA* piBR181-glk.pgm2.galU.ugd.VvGT5 was fed with quercetin. One milliliter sample was drawn from each culture at 12 h time intervals until 72 h. The samples were centrifuged, and supernatants were extracted with a double volume of ethyl acetate, followed by drying the samples. Then, the samples were dissolved in 100 μL methanol. These reaction mixtures were analyzed by HPLC-PDA and LC-MS for confirmation of glucuronic acid conjugated derivatives of quercetin and apigenin.

## 2.5. Product Analysis and Quantification

Reverse-phase HPLC-PDA analysis was performed with $C_{18}$ column (Mightysil RP-18 GP (4.6 × 250 mm, 5 μm) connected to a photo-diode array (PDA) (spectrum acquisition range: 190–800 nm) using binary conditions of $H_2O$ (0.05% trifluroacetic acid buffer) and 100% acetonitrile (ACN) at a flow rate of 1 mL/min for 30 min. The ACN concentrations were; 10% (0–2 min), 70% (20–24 min), 100% (24–28 min), 50% (28–30 min). For quantification of flavonoids, calibration curves of authentic apigenin and quercetin were made using 10, 25, 50, 75, and 100 μg/mL concentrations. The exact mass of the compounds was analyzed using LC-QTOF-ESI/MS (ACQUITY (ultra-pressure liquid chromatography (UPLC), Waters, Mil-ford, MA, USA)-SYNAPT G2-S (Waters)) in the positive ion mode.

## 3. Results and Discussion

### 3.1. Selection of Plant UGTs and Their Activities

Flavonoids are an important group of plant secondary metabolites and are usually present in the human diet in the form of glycosides. Glycosylation typically occurs in the final step

of flavonoid biosynthesis, which is catalyzed by several UGTs displaying high substrate stereo- and regio-selectivity [24]. In this context, although plants could be considered to have multiple glycosyltransferases enzymes capable of conjugating diverse sugars to different classes of plant molecules, we cannot escape from the difficulties using them in microbial cell factory, as most of the eukaryotic enzymes are difficult to express functionally. Previously reported regiospecific and well-characterized plant glucuronosyltransferases (UGT88D8 and VvGT5) [25,26] were selected to synthesize valuable glucuronic acid conjugated flavonoids in an industrially important bacterial strain *E. coli*. These UGTs were codon optimized prior to synthesize based on the *E. coli* codon usage to facilitate functional expression and used in biotransformation. According to a previous report, UGT88D8 favored the substrate apigenin catalyzing glucuronidation at 7-OH position where as VvGT5 favored quercetin and conjugate glucuronic acid at 3-OH position. Thus, we tested these GTs with respective substrates. The HPLC-PDA and LC-MS analyses could easily detect the generation of apigenin 7-*O*-glucuronoid (scutellarin A) and quercetin 3-*O*-glucuronoid (miquelianin). Furthermore, we tested similar class of flavonoids (baicalein, luteolin, crysin) in vivo bio-transformation catalyzed by UGT88D8 and VvGT5 with myricetin, kampferol, fisetin, as substrates. When carrying out analyses in HPLC-PDA and LC-MS, we were unable to detect product peak(s) and unable to detect in mass (data not shown). This also supported that the aforementioned UGTs are regio-, stereo- as well as a substrate specific.

### 3.2. Cloning and Expression of NDP-Sugar Pathway Genes and Assembly in a Single Vector

The UDP-glucuronic acid pathway genes were responsible for increasing the desired sugar pool in the cell. Glucokinase (Glk) catalyzes to form glucose 6-phosphate (G-6-P) from primary carbon source D-glucose, phosphoglucomutase (Pgm2) was responsible for converting the G-6-P into glucose 1-phosphate (G-1-P), which was further converted into UDP-glucose by the enzyme glucose 1-phosphate uridylyltransferase (GalU). Finally, UDP-glucose 6-dehydrogenase (Ugd) catalyzes to convert UDP-glucose into UDP-glucuronic acid (Figure 1). These genes were individually cloned into the piBR181 vector and expressed as explained in materials and methods. Expression of soluble proteins was examined using SDS-PAGE analyses, comparing with the standard sized protein marker (data not shown). Ultimately, these genes were assembled into a single vector piBR181 and the recombinant plasmid was later transformed into *E. coli* hosts for the biotransformation process.

After successful expression of each protein in soluble form, these genes, including UGT88D8 and VvGT5, were cloned to construct single recombinant plasmids (Figure 2). Under the hypothesis that overexpression of each glycolysis gene should enhance the nucleotide diphosphate (NDP)-sugar pool in the cytosol, this vector was transformed into a bacterial platform for the biocatalysis reaction.

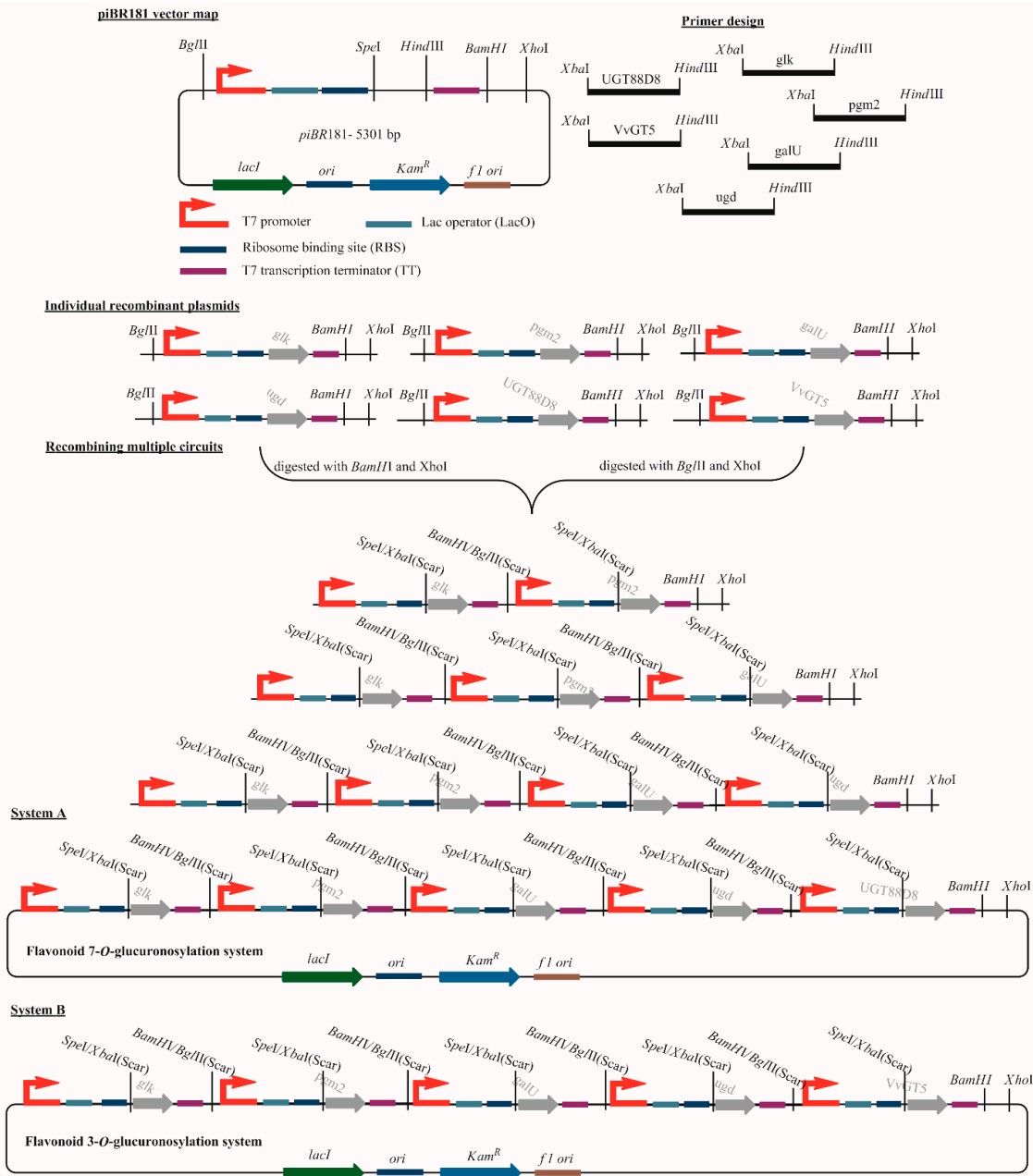

**Figure 2.** Schematic representation of the construction of single recombinant vector with the assembly of UDP-glucuronic acid pathway genes in mono-cistronic fashion. The final constructs piBR181-glk.pgm2.galU.ugd.UGT88D8 (system A) and piBR181-glk.pgm2.galU.ugd.VvGT5 (system B) contain sugar pathway genes; glucokinase (*glk*), phosphoglucomutase (*pgm2*), glucose 1-phosphate uridylyltransferase (*galU*), UDP-D-glucose dehydrogenase (*ugd*), flavonoid 3-*O*-glucuronosyltransferase (*VvGT5*) and flavonoid 7-*O*-glucuronosyltransferase (*UGT88D8*).

### 3.3. Synthesis of Glucuronoids in Recombinant Strains

The codon optimized UGTs from the respective source were initially screened by feeding in recombinant strains. *E. coli* BL21 (DE3)/piBR181-UGT88D8 and *E. coli* BL21 (DE3)/piBR181-VvGT5 were grown in LB medium, induced and incubated at 15 °C for 20 h to allow functional expression of the protein, as was explained in Materials and Methods. A total of 200 μM of apigenin and quercetin dissolved in dimethyl sulfoxide (DMSO) were added into the respective culture medium. After 48 h of incubation, 500 μL culture broth was taken and mixed with double volume ethyl acetate and vigorously vortexed followed by centrifugation to separate the organic and aqueous layer. The dried

supernatant organic layer was dissolved in 100 µL of methanol and subjected to HPLC analysis. New peak(s) were detected at retention time ($t_R$) ~ 17.5 min before the standard apigenin $t_R$ ~ 20.2 min. in the case of UGT88D8 at UV absorbance of 300 nm. While in VvGT5 culture sample, new peak(s) at $t_R$ ~ 17.5 were detected as compared to the standard quercetin $t_R$ ~ 19 min. These samples were further analyzed by high resolution mass analyses to confirm if the generated peaks were respective glucuronoids. The result was revealed by showing the mass spectrum of biocatalysis reaction mixture in positive mode as [apigenin 7-*O*-glucuronoid +H]$^+$ *m/z* 447.0911 (calculated mass 447.0927) and [quercetin 3-*O*-glucuronoid + H]$^+$ *m/z* 479.0821 (calculated mass 497.0826) (Figure 3). The conversion of apigenin to scutellarin A was ~6% while that of quercetin to miquelianin was ~13%. The conversion was unexpectedly low. Thus, we presumed that the low conversion was due to the low cytosolic pool of UDP-glucuronic acid.

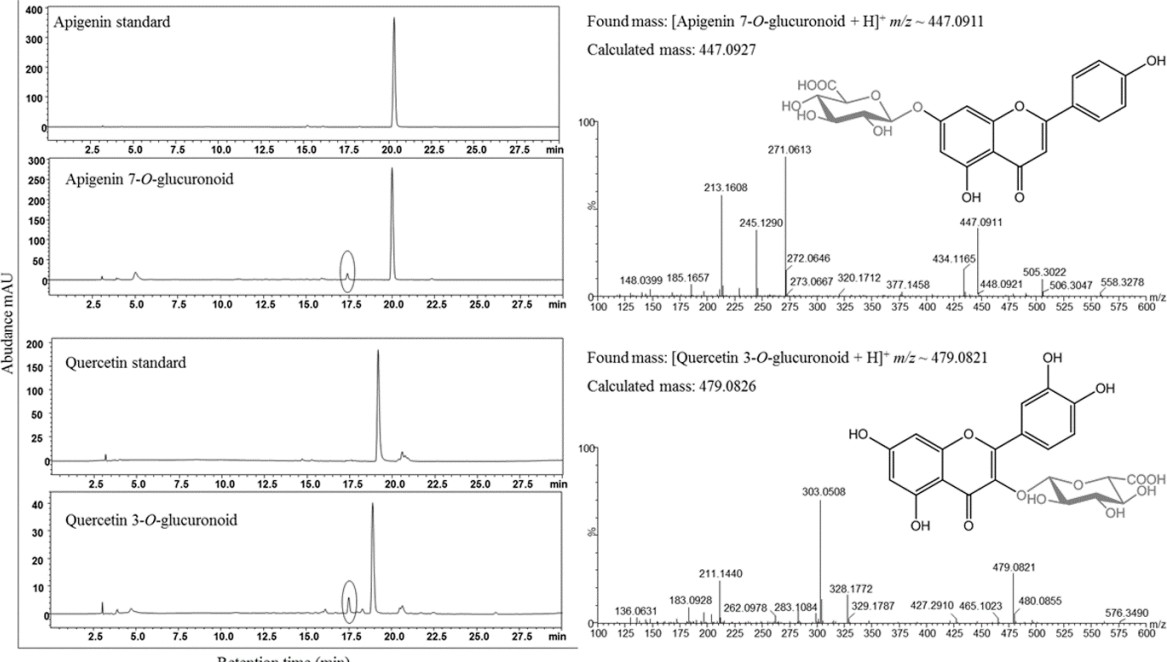

**Figure 3.** HPLC-PDA analysis of bio-catalysis reaction mixtures showing the generation of apigenin 7-*O*-glucuronoid (scutellarin A) and quercetin 3-*O*-glucuronoid (miquelianin) peak(s) by a circle in each chromatogram compared with their standard reference peak(s). Additionally, the mass spectrum of each reaction mixtures confirmed the target products.

*3.4. Enhanced Synthesis of Miquelianin and Scutellarin A*

After a successful confirmation of target products through spectroscopic analyses, we used other recombinants generated for this study. Two single recombinant vectors generated after assembly of UDP-glucuronic acid biosynthesis pathway genes and regiospecific UGTs were compared for increased production of respective products. Based on previous experiences, we hypothesized that the production of glycosides should increase upon overexpression of UDP-glucuronic acid pathway specific genes, along with *UGT88D8* and *VvGT5*. As in previous experiments, 50 mL cultures were grown and allowed the functional expression of protein for 20 h at 15 °C. For biotransformation reaction, the same 200 µmol/L concentration of substrates apigenin and quercetin were supplemented in respective recombinant strains. The cultures were harvested in 48 h incubation and analysed by HPLC-PDA. The production of each glycoside was increased approximately twice. The conversion of apigenin to scutellarin A was ~14% (28 µmol/L; 12 mg/L) and that of quercetin to miquelianin was ~31% (62 µmol/L; 30 mg/L). However, the increment was not satisfactory. We optimized the experiments with the addition of glucose as a carbon source. The production did not change significantly, although the cell growth was reasonably high (Figure 4).

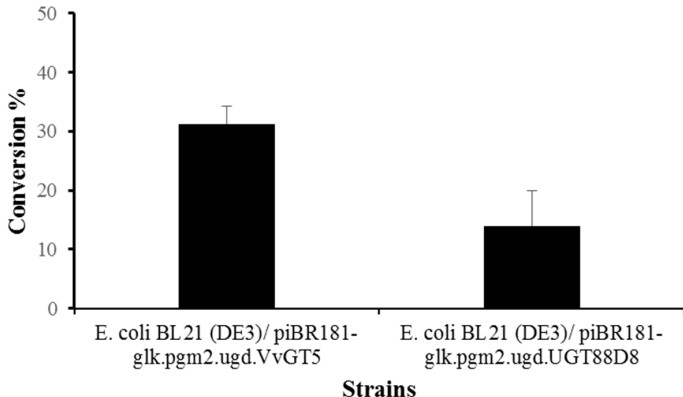

**Figure 4.** Production of meliquinnin and sculaterrin A using strain *E. coli* BL21 (DE3) harboring piBR181-glk.pgm2.galU.ugd.UGT88D8 and piBR181-glk.pgm2.galU.ugd.VvGT5.

Several plant metabolites, including flavonoids and their glycoside derivatives, have been produced from engineered *E. coli* cells as a microbial cell factory. Such microbial production has attracted extensive interest in the development of a novel system for the modification and efficient production of highly useful value-added compounds [14]. To produce glycosylated flavonoids from engineered *E. coli*, the pool of activated NDP-sugars, the donor substrates of UGTs and limiting factor in the cell cytosol is enhanced by introducing heterologous genes encoding the target sugar pathway genes, or by blocking the indigenous competing genes [14]. Using this approach, several flavonoid glycosides are produced from *E. coli*. For example, 3-*O*-rhamosyl quercetin, 3-*O*-rhamnosyl kaempferol, and 3-*O*-allosyl quercetin [27], astragalin [28], quercetin-3-*O*-4-deoxy-4-formamido-L-arabinose [29], quercetin 3-*O*-*N*-acetylglucosamine [30], quercetin-3-*O* xyloside [31] and quercetin 3-*O*-arabinoside [32], flavonol 3-*O*-rhamnosides [33,34], flavone 6-C-glucosides [35] are some representative flavonoid glycosides produced from *E. coli*. Using a similar approach, several polyketide glycosides have been produced from engineered microbial hosts [36]. Most of these systems used multiple vectors for overexpression of genes, which required the addition of multiple antibiotics to the culture broth for selection pressure. Thus, to reduce the antibiotic stress on the growing cell, we used a single vector system in which multiple genes were assembled for biosynthesis of two bioactive molecules; miquelianin and scutellarin A. Previously, miquelianin was produced using an approach similar to the approach that produced engineered *E. coli* cell [37]. However, this is the first report to produce scutellarin A from *E. coli* system. The production of miquelianin and scutellarin A from the systems developed in this study is relatively low compared to previous studies that reported biosynthesis of flavonoid glycosides [14]. However, this study opens possible paths for further harnessing of systems and synthetic biology tools to engineer central carbon metabolic pathways, to dam up the flow of UDP-glucuronic acid, and to develop an efficient system to produce glucuronic acid conjugated flavonoids.

## 4. Conclusions

Two different recombinant plasmids were developed and successively incorporated into the industrially important bacterium *E. coli*. The recombinant strains were used to synthesize glucuronoids of valuable dietary flavonoids. We successfully produced scutellarin A and miquelianin in engineered *E. coli* strains. A further experiment was carried out to construct a vector by assembling entire sugar biosynthesis pathway genes, aiming to increase the pool of UDP-glucuronic acid in the cytosol considering that pool of UDP-glucuronic acid is a limiting factor for lower production of respective compounds. However, when we used the final recombinant strains for biotransformation of quercetin and apigenin, we were unable to achieve high yield from the culture medium. Thus, there are two possible reasons for low production of target compounds: (1) the catalytic activity of synthesized UGTs (UGT88D8 and VvGT5) were not sufficient enough to conjugate sugar moiety, or (2) the downstream

pathway genes of UDP-glucuronic acid are actively utilizing UDP-glucuronic acid, thus limiting the pool of UDP-glucuronic acid for UGTs. Importantly, further strategic engineering of host bacterium or troubleshooting the problems could achieve an appreciable yield of desire glycosides.

**Author Contributions:** R.P.P. and P.P. conceived the idea. H.Y.J., P.P. and R.P.P. performed experiments, T.H.T.N. and P.B. assisted in recombinant construction. P.P., R.P.P. and J.K.S. wrote the manuscript. R.P.P. and H.Y.J. contributed equally to this work.

**Funding:** This research was funded by Next-Generation BioGreen 21 Program grant number [SSAC, grant#: PJ013137] Rural Development Administration, Republic of Korea.

**Conflicts of Interest:** The authors declare no conflicts of interest.

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
