# Peer review of "A Synthetic Approach for Biosynthesis of Miquelianin and Scutellarin A in Escherichia coli"

_applsci, doi:10.3390/app9020215_

Round 1

Reviewer 1 Report

The manuscript by Pandey et al. describes a synthetic approach for the biosynthesis of Miquelianin and Scutellarin A in E. coli. The topic is definitely interesting considering its potential application in the synthesis of natural products of enormous benefit to human health. The manuscript is organized and sufficiently describes the experimental procedure, yet it will benefit from additional editing as some sentences are poorly constructed.

Furthermore, in page 2; line 55-56: The author states that “Thus, conjugation of such sugar in promising dietary products should enhance its present potency.” What literature supports this statement? In fact, glucuronidation is Phase II metabolism in human that generally compromises the pharmacological activity of many drugs and enhances their biliary (and to some extent) renal elimination. In other words, glucuronidation indeed decreases the therapeutic value of many molecules.

The authors are advised to add a short paragraph about the other strategies by which the two products (Miquelianin and Scutellarin A) have been synthesized. It would be also interesting to add a paragraph about other natural products that have been synthesized by the strategy reported in this manuscript.  

Author Response

Reviewer #1

The manuscript by Pandey et al. describes a synthetic approach for the biosynthesis of Miquelianin and Scutellarin A in E. coli. The topic is definitely interesting considering its potential application in the synthesis of natural products of enormous benefit to human health. The manuscript is organized and sufficiently describes the experimental procedure, yet it will benefit from additional editing as some sentences are poorly constructed.

Response: Thank you so much for going through the manuscript and finding research interesting. We also thank the reviewer for providing suggestions for improvement of the manuscript.

Furthermore, in page 2; line 55-56: The author states that “Thus, conjugation of such sugar in promising dietary products should enhance its present potency.” What literature supports this statement? In fact, glucuronidation is Phase II metabolism in human that generally compromises the pharmacological activity of many drugs and enhances their biliary (and to some extent) renal elimination. In other words, glucuronidation indeed decreases the therapeutic value of many molecules.

Response: We removed this sentence from the manuscript not to make readers confused.

The authors are advised to add a short paragraph about the other strategies by which the two products (Miquelianin and Scutellarin A) have been synthesized. It would be also interesting to add a paragraph about other natural products that have been synthesized by the strategy reported in this manuscript.

Response: A paragraph is added just before the conclusion (L254-277).

Reviewer 2 Report

Dear Sir or Madam,

the manuscript „A Synthetic Approach for Biosynthesis of 2 Miquelianin and Scutellarin A in Escherichia coli “ describes construction a single vector system to overexpress entire UDP-glucuronic acid biosynthesis pathway genes in Escherichia coli BL21 (DE3). It can be published, but some problems with the manuscript need to be fixed before

Major remarks:

1.      English requires intensive correction. Specifically, articles are used wrong, or not used at all, where is appropriate.

2.      Use only past tense when describing methods and results

3.      Line 137 and everywhere as applicable: please, use SI units for concentration – replace M with mol/L.

4.      Although you have a list of abbreviations, all abbreviations need to be explained by the first use. For example, it is done in lines 140 – 145. Just make so everywhere please.

5.      THE MAIN PROBLEM! Where is discussion?

Minor remarks:

1.      Lines 14, 17 and everywhere as applicable: species names need to be given in italic

2.      Line 23: add “a” – “…a good start…”

3.      Paragraph ll. 47 – 56: I would suggest finishing this paragraph with a sentence introducing synthesis of flavonoid conjugate as prospective food additives – this was the idea, wasn’t it?

4.      Line 98: “sugar pathway genes” – specify, what sugar pathway you mean here

5.      Line 99: when you refer to a German research institute, it is good to say, what society it is – Max Planck, Leibniz, Helmholtz or Fraunhofer. And specify city please.

6.      Line 140: “The flasks were then incubated at 15 oC and 200 rpm for 60-72 h” – you mean here, to reach target OD? Then put it in the appropriate section, please.

7.      Lines 143 – 145 and 163: the name of the technique is wrong: it is LC-ESI-QqTOF-MS, i.e. liquid chromatography-electrospray ionization-quadrupole-time of flight mass spectrometry

8.      Line 154: the correct abbreviation for the technique is LC-MS, but not LC/MS. And, confirmation of what you mean here?

9.      Line 158: what does “PDA (300)” mean? This technology is designed for spectrum acquisition… It is confusing – change please.

10.  Line 168: what is “post modification”?

11.  Lines 187 – 192: does it really belongs to results?

12.  Line 218: do you mean “absorbance at 300 nm”?

13.  Line 222: What does “m/z ~447.0911” mean? Four decimal is precise enough – remove “~”

14.  Line 224: How was the % conversion calculated?

Author Response

Reviewer#2

The manuscript “A Synthetic Approach for Biosynthesis of 2 Miquelianin and Scutellarin A in Escherichia coli” describes construction a single vector system to overexpress entire UDP-glucuronic acid biosynthesis pathway genes in Escherichia coli BL21 (DE3). It can be published, but some problems with the manuscript need to be fixed before

Major remarks:

1.            English requires intensive correction. Specifically, articles are used wrong, or not used at all, where is appropriate.

Response: The manuscript is English revised and corrected as the reviewer suggested.

2.            Use only past tense when describing methods and results.

Response: The manuscript is thoroughly revised as the reviewer suggested.

3.            Line 137 and everywhere as applicable: please, use SI units for concentration – replace M with mol/L.

Response: Corrected throughout the manuscript.

4.            Although you have a list of abbreviations, all abbreviations need to be explained by the first use. For example, it is done in lines 140 – 145. Just make so everywhere please.

Response: The corrections have been made throughout the manuscript.

5.            THE MAIN PROBLEM! Where is discussion?

Response: A new paragraph just before the conclusion section is added as a part of the overall discussion (L254-277) with suitable references [27-37]. Since we do not have a separate discussion section, the discussion of the result is included within the results and discussion part.

Minor remarks:

1.            Lines 14, 17 and everywhere as applicable: species names need to be given in italic

Response: The scientific names including species are italicized.

2.            Line 23: add “a” – “…a good start…”

Response: Change has been made.

3.            Paragraph ll. 47 – 56: I would suggest finishing this paragraph with a sentence introducing synthesis of flavonoid conjugate as prospective food additives – this was the idea, wasn’t it?

Response: The sentence is added in the end of the paragraph.

4.            Line 98: “sugar pathway genes” – specify, what sugar pathway you mean here

Response: The information is added. UDP-glucuronic acid sugar pathway genes.

5.            Line 99: when you refer to a German research institute, it is good to say, what society it is – Max Planck, Leibniz, Helmholtz or Fraunhofer. And specify city please.

Response: The information is added in the respective place.

6.            Line 140: “The flasks were then incubated at 15 oC and 200 rpm for 60-72 h” – you mean here, to reach target OD? Then put it in the appropriate section, please.

Response: No. the flasks were kept at 15oC for 60-72 h for biotransformation. The sentence is corrected at the respective place.

7.            Lines 143 – 145 and 163: the name of the technique is wrong: it is LC-ESI-QqTOF-MS, i.e. liquid chromatography-electrospray ionization-quadrupole-time of flight mass spectrometry.

Response: Correction was made in both places.

8.            Line 154: the correct abbreviation for the technique is LC-MS, but not LC/MS. And, confirmation of what you mean here?

Response: Correction has been made throughout the manuscript.

9.            Line 158: what does “PDA (300)” mean? This technology is designed for spectrum acquisition… It is confusing – change please.

Response: corrected.

10.        Line 168: what is “post modification”?

Response: The sentence is rearranged.

11.        Lines 187 – 192: does it really belongs to results?

Response: We agree with the reviewer’s point that this is not the obtained result from any experiment. However, we believe this part is essential to explain figure 1 which is connected to the cloning of genes and construction of sugar cassette. Thus, this part will give clear information to understand the reasons for the cloning of these genes.

12.        Line 218: do you mean “absorbance at 300 nm”?

Response: Yes. The correction has been made.

13.        Line 222: What does “m/z ~447.0911” mean? Four decimal is precise enough – remove “~”

Response: Corrections has been made.

14.        Line 224: How was the % conversion calculated?

Response: The conversion was calculated as follows:

%Conversion= [Area of product peak / (sum of the area of product peak and substrate peak) x 100]

Round 2

Reviewer 2 Report

The authors have corrected the manuscript as was proposed. I would suggest to accept the manuscript now.